# Biomolecular Minerals and Volcanic Glass Bio-Mimics to Control Adult Sand Flies, the Vector of Human *Leishmania* Protozoan Parasites

**DOI:** 10.3390/biom13081235

**Published:** 2023-08-10

**Authors:** Kaiying Chen, Jean Marcel Deguenon, Roger D. Lawrie, R. Michael Roe

**Affiliations:** Department of Entomology and Plant Pathology, North Carolina State University, 3230 Ligon Street, Raleigh, NC 27695, USA; kchen23@ncsu.edu (K.C.); jeandeguenon@gmail.com (J.M.D.); roger.lawrie@upr.edu (R.D.L.)

**Keywords:** industrial biomolecular minerals, sand flies, Imergard, Celite

## Abstract

Sand flies (Diptera: Psychodidae) serve as vectors for transmitting protozoan parasites, *Leishmania* spp., that cause the disease called leishmaniasis. The main approach to controlling sand flies is the use of chemical insecticides. The discovery of alternative methods for their control is needed because of potential health risks of chemical insecticides and development of sand fly resistance to these pesticides. The biomineral produced by diatoms (diatomaceous earth, DE; Celite) and a volcanic glass bio-mimic (Imergard) have been shown by our group to be efficacious against mosquitoes, filth flies, and ticks but never studied for the control of sand flies. In a modified World Health Organization cone test, 50% of adult *Phlebotomus papatasi* sand flies at 29 ± 1 °C, 55 ± 5% RH, and 12:12 LD, when exposed to Imergard and Celite, were dead in 13.08 and 7.57 h, respectively. Proof of concept was established for the use of these biominerals for sand fly and leishmaniasis disease control. Using a light source as an attractant to the minerals had no significant effect on the LT_50_, the time to 50% mortality. The LT_50_ at a higher relative humidity of 70 ± 5% increased to 20.91 and 20.56 h for Imergard and Celite, respectively, suggesting their mode of action was dehydration. Scanning electron microscopy of dead sand flies showed high coating levels of Celite only on the sides of the thorax and on the tarsi, suggesting an alternative mode of action for mechanical insecticides.

## 1. Introduction

Sand flies (Diptera: Psychodidae), mainly in the genus *Phlebotomus* and *Lutzomyia*, serve as vectors for transmitting protozoan parasites, *Leishmania* spp., in the family Trypanosomatidae, that cause the disease leishmaniasis [1]. The transmission of these parasites to humans occurs through the bite of an infected female sand fly. The three common forms of the disease are cutaneous, visceral, and mucocutaneous leishmaniasis [2]. Cutaneous leishmaniasis is characterized by the formation of skin lesions, which when numerous can cause permanent scarring and severe disability [3]. Visceral leishmaniasis affects multiple internal organs that can be fatal if untreated [4]. Mucocutaneous leishmaniasis results in partial or complete destruction of mucous membranes in the nasal, oral, and throat cavities and surrounding tissues [5]. Each form is species specific in the *Leishmania* genus. An estimated 1 billion people are at risk for contacting this disease, with annual incidence rates of 1 million cases of cutaneous leishmaniasis and 30,000 cases of visceral leishmaniasis [6].

The most common method used to control sand flies is chemical pesticide treatments, for example, insecticide-treated bed nets and indoor residual spraying [7,8]. However, the use of synthetic insecticides could have unintended consequences on non-target arthropods, could pose risks to human health, and would be less desirable to many people compared to a non-chemical, mechanical approach.

Diatomaceous earth (DE) is an inorganic biomolecule produced by diatoms, mined from earth deposits that have accumulated over eons of time. Some species like that in Celite 610 are insecticidal by a physical mode of action and have been used in insect management [9,10,11]. The application of DE to control vector important arthropods like sand flies is interesting because this approach has not been considered before and if successful could avoid the concerns associated with chemical pesticides. Also, the attempted development of other minerals that could mimic DE to control pest arthropods has largely been unsuccessful. Recently, we discovered a new mechanical insecticide called Imergard^TM^ that is derived from volcanic rock as a DE mimic that was efficacious against mosquitoes [12,13], filth flies [14], and ticks [15], but was not examined against sand flies. The proposed mechanism of action for Celite and Imergard involves the abrasion of the insect’s outer, cuticular layer through physical contact and/or the absorption of cuticular lipids, leading to dehydration and death [16].

The current study investigated the effectiveness of Celite and Imergard for the control of adult sand flies, *Phlebotomus papatasi*, under different environmental conditions. Research was also conducted to better understand the mode of action of these mechanical insecticides on sand flies. The long-term goal is to demonstrate proof of concept that DE and the biomolecular mimic of DE, Imergard, could be a new, non-chemical approach to adult sand fly control.

## 2. Materials and Methods

### 2.1. Sand Fly and Mechanical Insecticides

*Phlebotomus papatasi* larvae and pupae (stages L3 and pupae *P. papatasi*, Turkey strain PPTK, NR-44000) were obtained from the Walter Reed Army Institute of Research, BEI Resources, NIAID, NIH (American Type Culture Collection, 10,801 University Blvd., Manassas, VA, USA) and maintained through the adult stage under Level II biological containment (ACL-2, arthropod) in the Dearstyne Entomology Laboratory (3230 Ligon Street, North Carolina State University, Raleigh, NC, USA) by methods approved by NCSU Environmental Health and Public Safety (BUA #2019-10-818). Larval pots were placed in a plexiglass (30.48 × 30.48 × 30.48 cm) mosquito rearing cage [17], housed in a foldable butterfly rearing cage (Restcloud, Chengdu, China) in a walk-in incubator (maintained at 26 °C, 80% relative humidity, and 12:12 (L:D) cycle). The incubator was in a room in the Dearstyne Entomology Building, where the only access door was nominally closed and where the door opening into the room was covered with mosquito netting. Fly paper (two traps, Black Flag, distributed by the Chemsico Division of the United Industries Corporation, St. Louis, MO, USA) was deployed from the ceiling to kill any flies that might escape from the walk-in incubator. Access to the room with the walk-in incubator was restricted to only BSL II trained researchers, and the room was fitted with two Black Flag fly paper traps and a Model BZ-15N Black Flag Bug Zapper. No flies were ever detected outside of the walk-in incubator. The pots were opened twice a week to release adult flies into the mosquito-rearing cage (the first containment level outside of the pots). The pots were gently tapped to promote fly emergence from the soil substrate. The bottoms of the pots were soaked in 2.5 cm of glass-distilled water for 10 min each time to maintain an adequate moisture level. The mosquito cage contained a 50 mL plastic test tube held with the open end up, filled with a 30% sucrose solution in glass-distilled water and with a cotton wick projecting from the sucrose solution through the open end of the tube to provide food and water to the adult sand flies.

Mechanical insecticides, Imergard and Celite 610, were provided by the company Imerys (Imerys Filtration Minerals, Inc., Roswell, GA, USA). Imergard is made from volcanic rock and Celite is made from diatomaceous earth. Both products were maintained at room temperature and humidity in the dark in their original commercial packaging until used.

### 2.2. Bioassay

A modified WHO cone test device (Figure 1) was used to measure Imergard and Celite lethal activity against adult sand flies (mixed sexes). The device consists of a Petri dish bottom (13.8 cm diameter), glass funnel (with a 9.9 cm diameter large opening, a 1.6 cm diameter small opening, and a 12 cm height), orange stopper, and plastic cap [14]. Imergard or Celite was spread as evenly as possible on each petri dish bottom (Figure 1) using a clean metal spatula at the rate of 26.25 g/m^2^. Twenty 5–10 d old (after emergence) adult flies (mixed sexes) were then transferred by aspiration from the mosquito cage to the cone test device, where the cap was removed and the orange stopper lowered to the middle of the funnel. After transferring, the orange stopper was raised to the sealed position and the cap re-applied. The cone test devices were then placed on a flat surface in an incubator at 29 ± 1 °C, 55 ± 5% RH, and a 12:12 L:D photoperiod. The bioassay modification from the WHO cone test [18,19] included the use of the stopper (Figure 1, not in the WHO test) and the assay bottom positioned on a flat surface (versus at an angle to the flat surface in the WHO bioassay method). Control tests were not treated with mechanical insecticides and were conducted at the same time as the treatments. Four replicates were conducted on different days. Flies were considered dead when there was no fly movement when gently moving the cone test device back and forth on the horizontal surface supporting the Petri dish bottom. Mortality data were recorded every three hours. The tests were started during the early part of the photophase, and in some cases, mortality had to be assessed during the scotophase by briefly shining a light source into the cone test.

### 2.3. Bioassay with Light Source under the Petri Dish

In the bioassay described in Section 2.2, we observed the adult sand flies were standing mostly on the inside surface of the cone. This was not the case for similar studies conducted with mosquitoes [12] and with three species of filth flies [14]. Studies so far also suggested that mosquitoes, filth flies, and ticks were not significantly repelled by Celite or Imergard [12,13,14,15]. *P. papatasi* is known to be attracted to white light [20]. Therefore, we placed a light (Nebo Eye Smart Sensor Light, 260 Lumens; Washington, DC, USA) just below the Petri dish bottom to possibly increase sand fly interactions with Celite and Imergard. There was no increase in temperature from the use of the light. Twenty 5–10 d old adult flies were transferred as described earlier. The cone test devices were transferred to the incubator at 29 ± 1 °C, 55 ± 5% RH, and with the lights off in the incubator. Three replicates were conducted on different days. Control tests were not treated with mechanical insecticides and were conducted at the same time as the treatments. Observations were made as described in Section 2.2.

### 2.4. Bioassay at High Humidity

If dehydration is the mode of action of Celite and Imergard, increasing the humidity should increase the time to death. Twenty 5–10 d old adult flies were transferred as described earlier to each cone test device treated with Imergard or Celite (with no light source below the Petri dish). The cone test devices were then incubated at 29 ± 1 °C, 70 ± 5% RH, and a photoperiod of 12:12 (L:D). This experiment was repeated four times on different days. Control tests were not treated with mechanical insecticides and were conducted at the same time as the treatments.

### 2.5. Statistical Analysis

Time course data were analyzed by Probit analysis using SAS (Version 9.4, Cary, NC, USA) with mortality Abbott corrected [21]. The LT_50_ and LT_80_ are the times to 50% and 80% mortality, respectively, and were calculated using the Probit model with Abbott correction. When the 95% confidence limits of the LT_50_ and LT_80_ values do not overlap in comparisons, they are considered statistically significantly different.

### 2.6. Scanning Microscopy

To identify if mechanical insecticides had transferred from the treated surface to the sand flies in the modified cone test device, scanning electron microscopy (SEM) was used to visualize Celite on the surface of adult sand flies that had died in the cone test. Dead sand flies were transferred individually from the cone test bottom to a glass vial and sent to the Analytical Instrumentation Facility at North Carolina State University. After being vacuum dried for 48 h, sand flies were attached to an aluminum Hitachi SEM mount using super glue. After that, they were coated with a 70 nm gold-palladium mixture (60 Au/40 Pd) using Cressington sputter coating for 60 s. To capture images of Celite on the surface of the sand flies, they were examined using a Hitachi SU3900 (Hitachi, Ltd., Chiyoda City, Tokyo, Japan) variable pressure scanning electron microscope.

## 3. Results

### 3.1. Efficacy of Imergard and Celite against Sand Flies

In cone tests (Figure 1), adult sand fly (mixed sexes) mortality was first observed after 3 h of exposure to Celite and Imergard at 29 ± 1 °C, 55% RH, and a 12:12 L:D cycle (Figure 2). Percentage mortality reached 100, 93.75, and 18.75 percent at 30 h for Celite, Imergard, and the control, respectively, after which the test ended. A probit model for time versus Abbot corrected mortality was developed (Table 1) for the results in Figure 2 and used to calculate the time to 50% (LT_50_) and 80% (LT_80_) mortality. The LT_50_ for Imergard was longer at 13.08 h (12.00–14.21 h, the 95% confidence interval (CI)) than that for Celite (7.57 h, 6.87–8.25 h) (Table 1). The LT_80_ for Imergard was also longer (26.50 h, 23.77–30.31 h, 95% CI) as compared to Celite (13.30 h, 12.28–14.49 h). The LT_80_ was also significantly longer than the LT_50_ for both minerals tested (Table 1).

### 3.2. Effect of a Light Source on Sand Fly Mortality

A light source was placed underneath the Petri dishes in our modified WHO cone test, and the assays conducted with the overhead incubator lights off in an attempt to better attract the sand flies to Imergard and Celite on the inside surface of the Petri dish.

Mortality was first observed at 3 h for both Imergard and Celite at 29 ± 1 °C and 55 ± 5% RH (Figure 3). Percentage mortality reached 95, 88.3, and 18.3 percent at 30 h for Celite, Imergard, and the control, respectively, after which the test ended. The LT_50_ and LT_80_ for flies exposed to Imergard with the light source were 11.59 h (7.06–16.46 h, 95% CI) and 28.34 h (19.31–75.89 h, 95% CI), respectively (Table 1). There were no significant differences in the LT_50_ and LT_80_ between assays with and without the light source. When exposed to Celite with the added light source, the LT_50_ was 8.23 h (6.72–9.66 h, 95% CI), and the LT_80_ was 17.11 h (14.63–20.80 h, 95% CI). At the LT_50_, again, there was no difference with the addition of the light source between Imergard and Celite while at the LT_80_, the time to 80% mortality was longer with the light source compared with no light source (Table 1). While there was a statistically higher time to 80% mortality for both Imergard and Celite compared to the LT_50_, no differences were found between these minerals for the same end point, unlike what was found in the absence of the light source.

### 3.3. Effect of Humidity on Sand Fly Mortality against Minerals

Assays were also run under high humidity (29 ± 1 °C, 70 ± 5% RH and 12:12 L:D). Mortality first occurred at 3 h for Imergard and at 9 h for Celite, and the mortalities were 67.5 and 90%, respectively, at 30 h, after which the study was ended (Figure 4) and were similar to the ending times for studies at the lower humidity (Figure 2). The LT_50_ for flies exposed to Imergard at 70% RH was 22.16 h (18.92–27.34 h, 95% CI) (Table 1). The LT_50_ for Celite was 20.82 h (16.93–27.81 h, 95% CI). There was no significant difference between Imergard versus Celite treatments under high humidity conditions at the LT_50_ (Table 1). The LT_80_ for Celite was 32.59 h (25.25–64.80 h, 95% CI). We were not able to calculate the LT_80_ for Imergard without extrapolation because the percent morality did not exceed 80% at 30 h. The LT_50_ for both Imergard and Celite was higher at 70% compared to 55% RH (Table 1).

### 3.4. Scanning Electron Microscopy

Scanning electron microscopy was conducted on dead sand flies after their exposure to Celite (Figure 5 and Figure 6). Three replicates were conducted, and the results shown are typical of all replicates. Only Celite was investigated because a US EPA registration for insect control is already granted, because of economy of time and resources, and because of limited access to instrumentation. The micrographs show Celite on the sand fly head, tarsi, wing, thorax, and abdomen. The amount of Celite in different parts of the sand flies varied. By far the highest coating levels were on the thorax (especially on the pleuron) and tarsi.

## 4. Discussion

Two industrial minerals, diatomaceous earth (DE, Celite 610) made by diatoms and a volcanic glass bio-mimic Imergard developed for mosquito and malaria control in Africa, were investigated for their insecticidal activity against unfed adult (mixed sexes) of sand flies, *Phlebotomus papatasi*. Phlebotomine sand flies are vectors of *Leishmania* protozoan parasites that cause human leishmaniasis, one of the most significant vector-borne diseases in the world. The insecticidal properties of minerals produced by diatoms, such as the species found in Celite, were known before. However, its use in controlling sand flies was never studied. In addition, we developed a mimic to DE called Imergard for mosquito control, which was more efficacious than Celite [12]. Imergard is derived from volcanic glass and is easily mixed in water and applied with conventional spray equipment used in vector control; as a residual wall spray inside of homes in Africa, Imergard was efficacious by WHO standards, without significant loss of activity for 6 mo [12,13]. Imergard is abundantly available and is considered safe as the “white stuff found in toothpaste”.

In a modified WHO cone test in this paper, we found that the LT_50_ for sand flies exposed to Imergard at 29 ± 1 °C and 55 ± 5% RH was 13.08 h; when exposed to Celite, the LT_50_ was 7.57 h. Deguenon et al. [12] used the same cone test and found that the Imergard LT_50_ for mosquitoes was 4.96 h at 32 ± 1 °C and 60 ± 5% RH. Chen et al. [14] found that the LT_50_s for Imergard in the cone test for three species of filth flies were in the range of 3.2–7.1 h at 30 ± 1 °C and 50 ± 5% RH. Sand flies exposed to Imergard had a longer LT_50_ compared to mosquitoes and filth flies. However, our sand fly studies demonstrated similar susceptibility to DE with that of mosquitoes and filth flies.

In the filth fly studies conducted by Chen et al. [14], Imergard exhibited faster mortality rates against blow flies and house flies than against larger flesh flies. As insects increase in size (assuming similar body shapes), their volume increases cubically while their surface area increases by the square. Korunic [22] also reported that DE had a greater negative impact on insects with smaller body sizes. This could be attributed to their larger surface area to volume ratio, leading to a comparatively higher loss of water compared to larger insects. The hypothesized mode of action of DE is disruption of the water protective layer of the insect cuticle by abrasion and/or absorption of lipids leading to dehydration and resulting in death. Sand flies in comparison to the filth flies studied by Chen et al. [14] are much smaller. Adult house flies weigh 20.1 mg [14] and adult sand flies weigh 0.22–0.30 mg [23]. The house fly adult is 67–91 times larger by weight than a sand fly. Based on size alone, the hypothesis would be that Imergard and Celite would have a greater effect on sand flies (have a shorter LT_50_) than for the larger house fly. However, the opposite was discovered. *P. papatasi* are highly adapted to arid and periarid bioclimates with high temperatures and low relative humidity [24,25], and because of this adaptation they might have naturally occurring resistance mechanisms to prevent dehydration, which also increased their tolerance to Imergard. These possible resistance mechanisms that increased the time to the LT_50_ in sand flies are unknown.

The effectiveness of DE against insects was shown before by others to decrease when humidity increased [22]. At the higher humidity, the rate of water evaporation from the insect decreases. Chen et al. [14] also reported that the efficiency of Imergard decreased (the LT_50_ was longer) at a higher relative humidity for filth flies, and Deguenon et al. [12] found the same for malaria mosquito adults, *Anopheles gambiae*. In the current study, the LT_50_s for Celite and Imergard against sand flies were longer as well at the higher humidity (Table 1), consistent with the hypothesized mode of action for these minerals which is dehydration. Increased humidity may also affect the electrostatic transfer of the mineral to the sand fly. There are no data on the impact of changes in humidity on electrostatics in the cone test.

Based on the scanning electron microscopy (SEM) images, Celite was predominantly observed on the pleural region of the thorax and on the tarsi of sand flies (Figure 5 and Figure 6). There was mineral found on other areas of the body, but this was minimal by comparison. This is drastically different from what Chen et al. [14] found for house flies exposed to Imergard using the same modified WHO cone test used for sand fly exposures. The entire body of the house fly was coated with Imergard, and the fly appeared essentially as an “Imergard mummy”. The setae on the pleuron of sand flies were much shorter with a lower density compared to that on the thorax dorsum, abdomen, wings, and head. This is different from house flies that have less setae of a shorter length. The functional reason for the long setae in sand flies is not completely clear but based on our findings with Celite (Figure 5 and Figure 6), might be a mechanism to protect the sand fly from small sand particles and other debris (and from Celite) making contact to the body. This is further supported by the findings of Chen et al. [14] that house flies were heavily coated with Imergard while sand flies were not.

The sand fly coating results are informative, showing that mineral deposits mostly on the pleuron of the thorax and on the tarsi could have resulted in a lethal effect. The flies shown in Figure 5 and Figure 6 were collected after they had died, while in the control experiments conducted at the same time with no minerals in the WHO funnel test, control mortality was zero. It is interesting that the spiracle that provides oxygen and carbon dioxide exchange to the brain and mouthparts is found (one each) on the pleuron of the thorax. Richardson et al. [15] hypothesized from SEM studies that nymphs of the black-legged tick, *Ixodes scapularis*, died after dipping into Celite because they had the spiracular plate blocked by the mineral, and Cave et al. [26] found the minerals inside of the aeropyles of the spiracular plate in adult *I. scapularis* that died by walking onto a treated surface of Celite. These studies suggested that the minerals might affect respiration or disrupt the water retention mechanisms associated with respiration. At the same time, the presence of minerals on the thoracic pleural region was correlated with mortality, but a “cause and effect” was not proven.

House flies in the modified WHO cone test [14] presented as much stronger fliers than the sand flies. House flies were more active and repeatedly flew at a high speed into the mineral-treated surface of the bioassay arena. Sand flies most often were seen standing on the tapered side of the glass funnel. A light source under the assay arena was used in an attempt to enhance sand fly interactions with the treated Petri dish. Sand flies are known to be attracted to light [20]. However, no significant difference was observed on the time to the LT_50_ for both Imergard and Celite compared to assays conducted without light (Table 1). Richardson et al. [15] studied the efficacy of Imergard and Celite against unfed nymphal black-legged ticks and reported that Imergard exhibited slightly higher efficacy compared to Celite by dipping. In contrast, our findings indicated that sand flies exposed to Imergard had a longer LT_50_ compared to those exposed to Celite with no difference at high humidity (Table 1). Our results were different from the findings reported by Richardson et al. [15]. Factors that might increase the time to death for sand flies compared to the filth fly studies of Chen et al. [14] are differences in overall activity levels and flight dynamics, the ability for sand flies to rest on the glass sides of the funnel versus the treated bottom of the cone test, differences in insect electrostatics, and sand fly natural resistance to dehydration.

## 5. Conclusions

In conclusion, the biomolecular mineral produced by diatoms, Celite 610, and a volcanic glass mimic, Imergard, in a modified WHO cone test were lethal to adult sand flies, *Phlebotomus papatasi*, establishing as proof of concept that they could be used to control this pest and reduce human disease. Sand flies vector a protozoan that causes an important human disease, leishmaniasis. The time to 50% sand fly mortality was longer when exposed to Imergard and Celite at high relative humidity, consistent with the hypothesized mode of action for these minerals which is dehydration. Scanning electron microscopy of dead sand flies showed high coating levels of Celite found mostly on the sides of the thorax and on the tarsi, suggesting that water loss associated from the disruption of the outer cuticle water-proofing layer of the body in toto is not the mode of action. Alternatively, the minerals might be affecting the function of the water retention features of the respiratory system. Additional semi-field and field research is necessary to determine the practical use of Celite and Imergard in controlling sand flies.

## Figures and Tables

**Figure 1 biomolecules-13-01235-f001:**
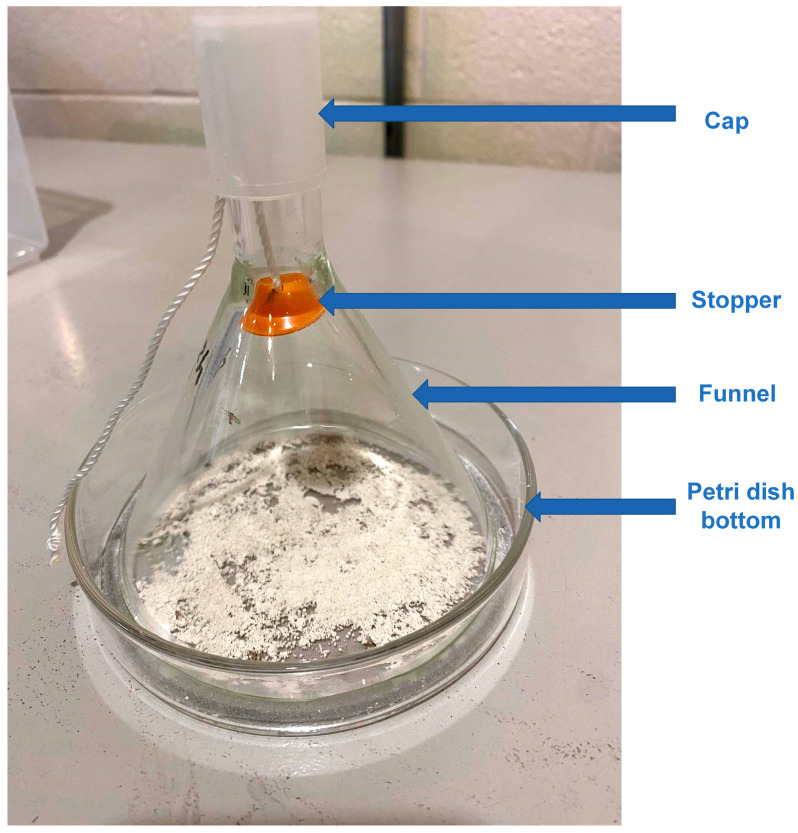
Architecture for the WHO modified cone test consisting of a cap, an orange stopper with a string, a funnel, and a Petri dish bottom. The white powder in the Petri dish bottom is Celite.

**Figure 2 biomolecules-13-01235-f002:**
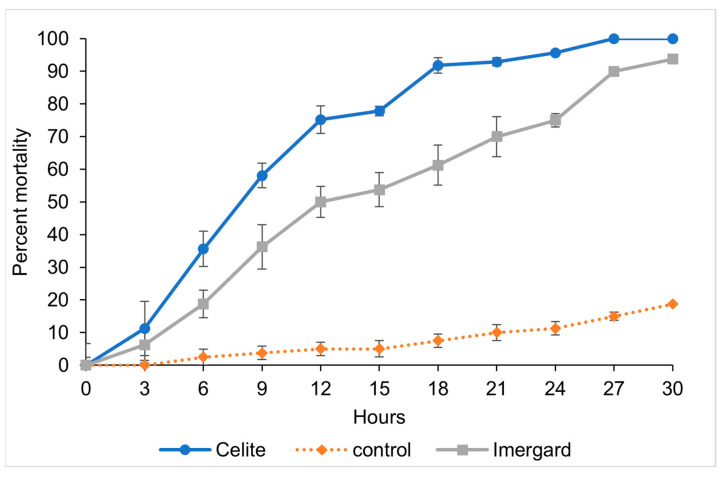
Time versus mortality for Imergard and Celite versus a negative control in a modified WHO cone test (Figure 1) at 29 ± 1 °C, 55 ± 5% RH and 12:12 L:D against mixed sexes of adult sand flies, *Phlebotomus papatasi*. The error bars are ±1 standard error of the mean, which in some cases were smaller than the size of the symbol.

**Figure 3 biomolecules-13-01235-f003:**
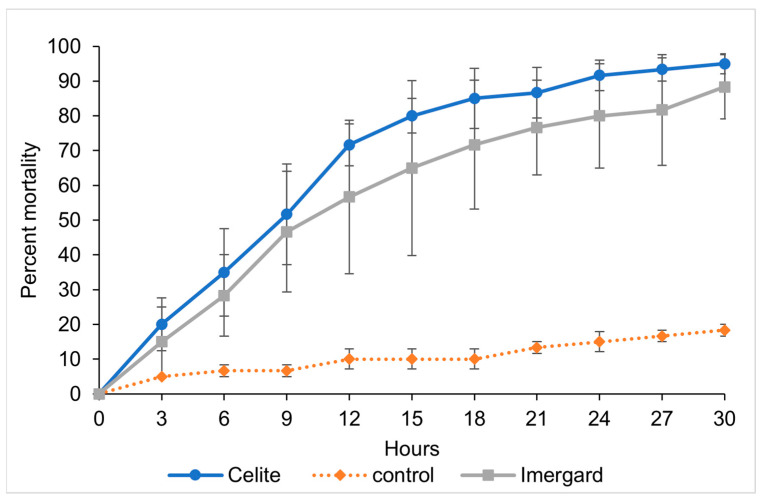
Time versus mortality for Imergard and Celite versus a negative control in a modified WHO cone test (Figure 1) at 29 ± 1 °C and 55 ± 5% RH (with the overhead incubator lights off and a light source below the glass Petri dish) against mixed sexes of adult sand flies, *Phlebotomus papatasi*. The error bars are ±1 standard error of the mean, which in some cases were smaller than the size of the symbol.

**Figure 4 biomolecules-13-01235-f004:**
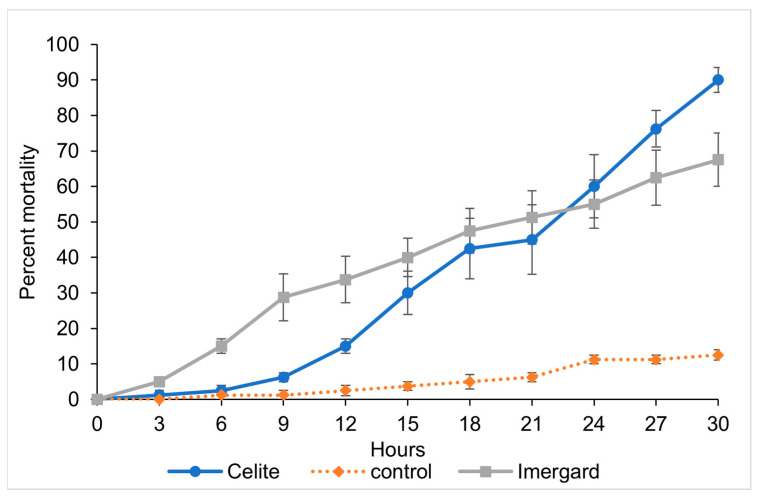
Time versus mortality for Imergard and Celite versus a negative control in a modified WHO cone test (Figure 1) at 29 ± 1 °C, 70 ± 5% RH, and a 12:12 L:D cycle against mixed sexes of adult sand flies, *Phlebotomus papatasi*. The error bars are ±1 standard error of the mean, which in some cases were smaller than the size of the symbol.

**Figure 5 biomolecules-13-01235-f005:**
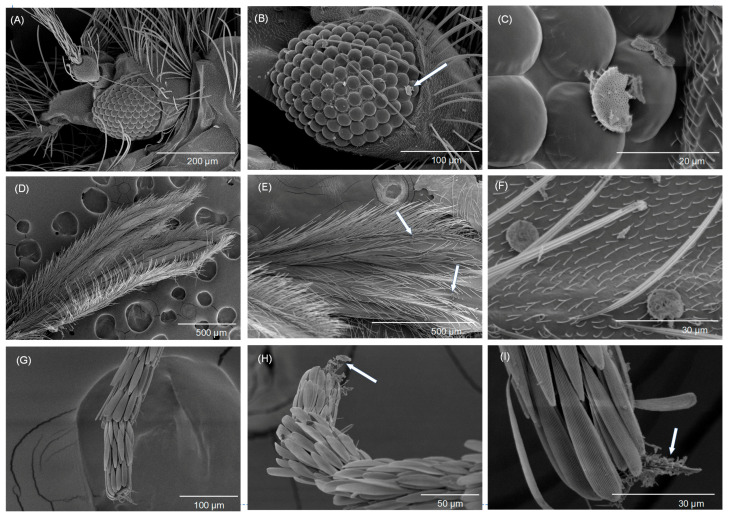
Scanning electron micrographs of an adult sand fly, *Phlebotomus papatasi*, exposed to Celite at the treatment rate of 26.25 g/m^2^. Cones were placed in an incubator at 29 ± 1 °C, 55 ± 5% RH, and 12:12 L:D. Dead flies were transferred to glass vials individually for imaging. (**A**) untreated head, (**B**) Celite treated head, (**C**) Celite treated head—higher magnification, (**D**) untreated wing, (**E**) Celite treated wing, (**F**) Celite treated wing—higher magnification, (**G**) untreated lower leg, (**H**) Celite treated lower leg, and (**I**) Celite treated lower leg—higher magnification. Arrows are pointing to Celite on the fly cuticle, which is shown at a higher magnification in the picture to the right.

**Figure 6 biomolecules-13-01235-f006:**
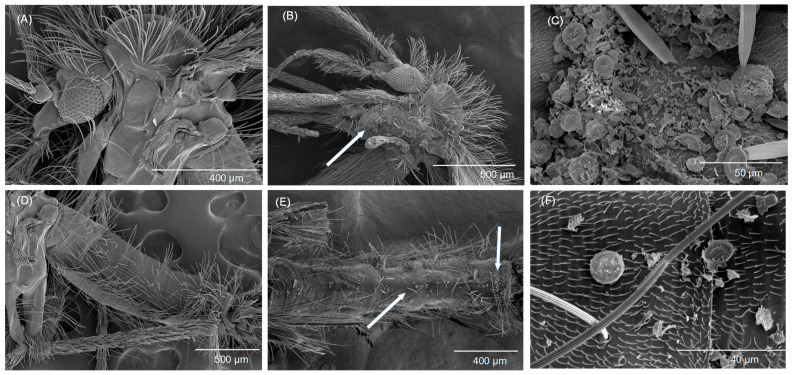
Scanning electron micrographs of an adult sand fly, *Phlebotomus papatasi*, exposed to Celite at the treatment rate of 26.25 g/m^2^ in the cone test. Cones were placed in an incubator at 29 ± 1 °C, 55 ± 5% RH, and 12:12 L:D. Dead flies were transferred to glass vials individually for imaging. (**A**) untreated thorax, (**B**) Celite treated thorax, (**C**) Celite treated thorax—higher magnification of pleuron of the thorax, (**D**) untreated abdomen, (**E**) Celite treated abdomen, and (**F**) Celite treated abdomen—higher magnification. Arrows are pointing to Celite on the fly cuticle, which is shown at a higher magnification in the picture to the right.

**Table 1 biomolecules-13-01235-t001:** Probit model and time to 50% (LT_50_) and 80% (LT_80_) mortality of adult sand flies, *Phlebotomus papatasi*, in a WHO modified cone test.

Mineral	Environment	*n*	Slope (SE) ^+^	LT_50_ (95% CL) ^++^	LT_80_ (95% CL) ^++^	χ^2 +++^
Imergard	29 ± 1 °C, 55% RH	80	2.75 (0.20)	13.08Aa (12.00–14.21)	26.50Ab (23.77–30.31)	43.65
Celite	29 ± 1 °C, 55% RH	80	3.44 (0.22)	7.57Ba (6.87–8.25)	13.30Bb (12.28–14.49)	44.35
Imergard	29 ± 1 °C, 55% RH, light	80	2.17 (0.58)	11.59ABa (7.06–16.46)	28.34ACDb (19.31–75.89)	205.64
Celite	29 ± 1 °C, 55% RH, light	80	2.65 (0.30)	8.23Ba (6.72–9.66)	17.11Cb (14.63–20.80)	51.62
Imergard	29 ± 1 °C, 70% RH	80	1.81 (0.23)	22.16C (18.92–27.34)	N/A	53.90
Celite	29 ± 1 °C, 70% RH	80	4.32 (1.13)	20.82Ca (16.93–27.81)	32.59ADa (25.25–64.80)	418.40

^+^ SE = standard error. ^++^ Results in hours, Abbott corrected. CL = confidence limit. In each column, different upper-case letters indicate no overlap of the 95% confidence intervals and were statistically significantly different. In each row, different lower-case letters indicated no overlap of the 95% confidence intervals and were statistically significantly different. ^+++^ Chi-square.

## Data Availability

Additional data not presented in this paper are available by request from the corresponding author.

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
