# Peer review of "Biomolecular Minerals and Volcanic Glass Bio-Mimics to Control Adult Sand Flies, the Vector of Human Leishmania Protozoan Parasites"

_biomolecules, 2023, doi:10.3390/biom13081235_

Round 1

Reviewer 1 Report

The manuscript by Chen et al. investigates the role of biominerals in sand fly and leishmaniasis disease control. Two biominerals, Celite and Imergard, were tested in a cone experiment to evaluate their ability to kill sand flies while also attempting to identify the compounds' mechanisms of action. Using this method, Chen et al. were able to confirm the significance of these mechanical insecticides in sand fly control and imply that dehydration was their route of action.

The paper is well written and the results depicted in figures are well presented and discussed. The conclusions are well supported by the presented results. I support the publication of this paper upon minor modifications.

Minor points:

In the abstract, abbreviations should be avoided as they will only be understood by very specialist people.

How was the amount of the celite and imergard (26.25 g/m2) determined?

Celite coating was only seen on the thorax and tarsi of dead sand flies, in contrast to what was described previously for filth flies. Could contact after death lead to celite coating? Can this be examined by allowing dead flies to contact celite for a couple of minutes? Contrary to mosquitoes or filth flies, sand flies don't always touch or explore celite, therefore they don't seem to be as impregnated with the mineral. Could swallowing of the biomaterial be the cause of death?

Why did the authors limit their investigation to celite deposition? Was the imergard deposition not looked into?

Lines 318-320 in discussion do not make sense, please correct this sentence.

Reviewer 2 Report

First of all, I would like to congratulate for the submitted study, addresses a pressing problem for the studied field, concerning insecticide use necessary for sand flies diseases control measures, and possible insecticide resistence onset in sand flies.

The study is well presented, a review of the English language is recommended, as well as the addition of a statistical comparison between the results.

Please consider to check throughout the whole text the spelling of sand fly or sand flies that must always be conventionally separated into two distinct words

A moderate review of the English language is recommended

Reviewer 3 Report

This manuscript describes the efficacy of two mechanical insecticides, Celite 610 and Imergard, over adult Phlebotomus papatasi (Diptera: psychodidae) sand flies, vector of Leishmaniasis.

Although the work is concise, the research methodology and results were expressed adequately. However, this reviewer thinks the discussion may be improved prior to publication in this Journal.

Authors can add a short comment on how they think it can be applied in the field since light didn’t pull insects to material (i.e. formulate with a lure, spray with wax, mix with flies food, etc) and/or cite works about what it is known about the impact over beneficial insects.

No doubt of the effect of the humidity on LT50. But if the conclusion suggested is dehydration, the discussion should be complemented. Can be discussed the incidence of %HR over the flies takeoffs, and the frequency of contact to Celite/Imergard. Same with the mobility behaviour of treated vs untreated insects. Non treatment insects have no option to hydration (water-soaked cotton). Are the weights of both substrates were tracked at different %HR. Also it is worth to comment the possibility of microparticles getting stuck to charged insects (DOI 10.1139/z62-051) by the triboelectric effect. The influence of humidity on this phenomena (DOI: 10.1039/C6SM02041K). It may be interesting to add an experiment using a static neutralizer gun or either a electroneutral net inside the cone to study the possible static transfer?

Other comments.

I would recommend checking the title owing to unaware readers can understand that biominerals can be used for chemotherapy to control the disease.

Line 38. If the estimation is from 2004 or before, I would recommend citing a recent report.

Line 97. Reference for Cone bioassay (from WHO guideline) should be herein. Modification introduced should be disclosed in Methods.

Line 343. flies "died slower” sounds confusing. 

Line 397. Should be “feeding”.

Round 2

Reviewer 2 Report

Title is not very clear in this new version, please consider to change it again.

In conclusions lines 373-375: "One approach for controling sand flies in the field would be spraying as a suspension in water or applied as a powder with a drop or broadcast spreader on the ground where the flies propagate", consider removing the sentence, it doesn't seem related to sand flies biology.

Double check editing of English language
